# Integrating RPA-LFD and TaqMan qPCR for Rapid On-Site Screening and Accurate Laboratory Identification of *Coilia brachygnathus* and *Coilia nasus* in the Yangtze River

**DOI:** 10.3390/foods14203484

**Published:** 2025-10-13

**Authors:** Yu Lin, Suyan Wang, Min Zhang, Na Wang, Hongli Jing, Jizhou Lv, Shaoqiang Wu

**Affiliations:** 1Chinese Academy of Quality and Inspection & Testing, Beijing 100176, China; linyu.work@foxmail.com (Y.L.); jinghl@caiq.orb.cn (H.J.); 2Center for Biosafety, Chinese Academy of Inspection and Quarantine, Sanya 572024, China; 3Technology Innovation Center of Animal and Plant Product Quality, Safety and Control, State Administration for Market Regulation, Beijing 100176, China

**Keywords:** *Coilia brachygnathus*, *Coilia nasus*, rapid species identification, RPA-LFD, TaqMan qPCR

## Abstract

Accurate differentiation between *Coilia brachygnathus* and *Coilia nasus* is imperative for the effective management of fisheries, the conservation of aquatic ecosystems, and the mitigation of commercial fraud. Current morphological identification remains challenging due to their high morphological similarity—particularly for processed samples—while conventional molecular methods often lack the speed or specificity required for field applications or high-throughput screening. In this study, a novel integrated approach was developed and validated, combining TaqMan quantitative real-time PCR (qPCR). for precise genotyping of *C. brachygnathus* and *C. nasus* with Recombinase Polymerase Amplification coupled with Lateral Flow Dipstick (RPA-LFD) for rapid on-site screening. First, species-specific RPA-LFD assays were designed to target the mitochondrial COI gene sequence. This enabled visual detection within 10 min at 37 °C, with a sensitivity of 10^2^ copies/μL, and required no complex equipment. A dual TaqMan MGB qPCR assay was further developed by validating stable differentiating SNPs (chr21:3798155, C/T) between *C. brachygnathus* and *C. nasus*, using FAM/VIC dual-labeled MGB probes. Results showed that this assay could distinguish the two species in a single tube: for *C. brachygnathus*, Ct values in the FAM channel were significantly earlier than those in the VIC channel (ΔCt ≥ 1), with a FAM detection limit of 125 copies/reaction; for *C. nasus*, only VIC channel amplification was observed, with a detection limit as low as 12.5 copies/reaction. Validation with 171 known tissue samples demonstrated 100% concordance with expected species identities. This integrated approach effectively combines the high accuracy and quantitative capacity of TaqMan qPCR for confirmatory laboratory genotyping with the speed, simplicity, and portability of RPA-LFD for initial field or point-of-need screening. This reliable, efficient, and user-friendly technique provides a powerful tool for resource management, biodiversity monitoring, and ensuring the authenticity of high-quality *C. brachygnathus* and *C. nasus*.

## 1. Introduction

The genus *Coilia* (grenadier anchovies) comprises thirteen species distributed primarily in the western Pacific and Indian oceans [1]. Within this genus, *Coilia nasus* is a highly valuable migratory fish species in East Asian fisheries, renowned for its nutritional quality and commercial significance—particularly in the main Yangtze River basin of China [2,3]. In contrast, *Coilia brachygnathus* (Yangtse grenadier anchovy) is a resident freshwater species primarily inhabiting the middle reaches of the Yangtze River basin, including its tributaries and connected lakes (e.g., Dongting, Poyang, and Liangzi Lakes). Morphologically, *C. brachygnathus* is distinguished from *C. nasus* by its shorter maxilla and fewer lateral longitudinal scales [4,5]. These two species exhibit significant divergence in morphology, physiology, and behavior, with *C. brachygnathus* demonstrating adaptations consistent with rapid phenotypic evolution in freshwater environments [6,7]. However, populations of *C. nasus* have declined severely due to overfishing, habitat degradation, and river fragmentation, leading to its designation as Endangered (EN) on China’s Biodiversity Red List [8,9].

To alleviate fishing pressure on C. nasus, fisheries management authorities have enacted a series of targeted measures; relevant fisheries management authorities have implemented a series of specialized regulatory measures [10]. These include strictly controlling the number of fishing licenses issued, significantly shortening the permitted fishing duration, and establishing seasonal fishing moratoriums. These measures are designed to directly reduce fishing intensity, promote the natural recovery of wild populations, and strike a balance between ensuring market supply and enforcing species conservation. Against this backdrop, accurate species identification is particularly crucial. It serves as the foundation for scientifically informed fisheries management and the maintenance of population resources, while also helping to protect consumers from mislabeling and commercial fraud, thereby upholding market fairness and integrity. Consequently, accurate species discrimination is critical for sustaining stocks through evidence-based fisheries management, protecting consumers from economic deception, and ensuring market integrity. These imperatives align with China’s prioritization of endangered aquatic species conservation and aquatic ecosystem restoration as key national ecological governance objectives.

Conventional morphological identification of *C. nasus* and *C. brachygnathus* is notoriously challenging due to their high phenotypic similarity, particularly during juvenile stages or when specimens are processed (e.g., frozen, dried, or canned). Molecular techniques offer a more reliable alternative. Methods such as conventional PCR [11], subsequent sequencing [12], multiplex PCR [13,14], and DNA barcoding (specifically cytochrome c oxidase subunit I) [15], have been employed for species authentication. However, these approaches typically require sophisticated laboratory infrastructure, skilled personnel, and prolonged turnaround times (often several hours to days), and are impractical for rapid on-site screening or large-scale monitoring at landing sites, markets, or border inspection points.

Isothermal nucleic acid amplification technologies offer a promising solution for field-deployable diagnostics. Among these, recombinase polymerase amplification (RPA) enables target sequence amplification within 10–20 min at 37–42 °C without the need for thermal cyclers, and is compatible with lateral flow dipstick (LFD) visualization [16]. RPA has been successfully deployed in field scenarios including animal disease surveillance and plant quarantine. For instance, its successful deployment in field detection of African swine fever virus (ASFV) confirms its field applicability (detection limit: 10 copies/μL within 15 min) [17]. In plant pathogen detection, RPA-LFD achieved a sensitivity of 0.0005 pg/μL for *Pantoea stewartii* subsp. Stewartii [18]. Notably, a fluorescent RT-RPA assay was recently developed for rapid diagnosis of viral hemorrhagic septicemia in fish [19], highlighting the technique’s potential in aquatic contexts. Despite these advances, RPA-LFD remains untested for aquatic species identification, and its ability to discriminate for highly similar species may occasionally require validation by higher-resolution methods.

In contrast, TaqMan probe-based quantitative real-time PCR (qPCR) offers exceptional specificity, sensitivity, and quantitative capability through the use of labeled hydrolysis probes. This minimizes false positives and enables precise species identification. Despite its high accuracy, TaqMan qPCR relies on thermal cycling instruments and thus remains confined to laboratory environments.

To bridge this technological gap, we propose a novel, integrated strategy that leverages the complementary strengths of both platforms: (1) RPA-LFD for primary rapid screening: It provides immediate, on-site differentiation capability for field personnel or inspectors without access to central laboratories. (2) TaqMan qPCR for confirmatory genotyping: It delivers high-precision, unambiguous species identification in the laboratory for samples flagged by RPA-LFD or requiring definitive legal/commercial validation.

In this study, we report the development and rigorous analytical validation of species-specific RPA-LFD and TaqMan qPCR assays targeting key genetic markers in *C. brachygnathus* and *C. nasus*. To our knowledge, this study establishes the first integrated framework that combines field-deployable RPA-LFD with laboratory-based TaqMan qPCR for discriminating against these ecologically and commercially vital species. Our dual-platform approach creates a versatile, scalable identification system applicable to diverse scenarios—from rapid port-side inspections to regulatory enforcement and biodiversity monitoring. By enabling accurate species authentication, this methodology directly supports conservation efforts in the Yangtze River ecosystem and combats fraudulent mislabeling in commercial fisheries.

## 2. Materials and Methods

### 2.1. Sample Collection

This study focused on *Coilia nasus*, *Coilia brachygnathus*, and co-occurring non-target fish species. From 2023 to 2025, we systematically collected specimens across five provinces and municipalities (Jiangxi, Anhui, Jiangsu, Zhejiang, and Shanghai) within the lower Yangtze River basin. A total of 171 whole fish specimens were obtained, comprising 29 *C. nasus*, 6 *C. brachygnathus*, and 136 individuals from closely related non-target species (Table 1). To ensure representative genetic diversity, a minimum of six individuals were sampled per species. All collections were conducted under fishery licenses authorized by the relevant provincial/municipal agricultural and rural affairs departments. Following initial morphological identification, dorsal muscle tissue was excised from each specimen and preserved at −80 °C for subsequent DNA extraction.

### 2.2. Genomic DNA Extraction

Genomic DNA was extracted from muscle tissue using the DNeasy Blood & Tissue Kit (Qiagen, Hilden, Germany; Cat. No. 69504) according to the manufacturer’s protocol, with the following modifications. Briefly, fish muscle tissue was ground into a powder in liquid nitrogen, followed by the addition of an appropriate amount of lysis buffer, then proteinase K digestion and RNase treatment. Subsequent steps followed the standard kit protocol for DNA extraction. DNA concentration and purity (A260/A280 ratio) were assessed using a NanoDrop 2000 spectrophotometer (Thermo Fisher Scientific, Waltham, MA, USA). Only DNA extracts with an A260/A280 ratio between 1.8 and 2.0 were used as templates for subsequent PCR amplification and stored at −20 °C.

### 2.3. Primer and Probe Synthesis and SNP Verification

The mitochondrial genome sequence of *Coilia nasus* from the Yangtze River (GenBank accession: KJ710626.2) was retrieved from NCBI (https://www.ncbi.nlm.nih.gov/, accessed on 20 March 2024). Multiple sequence alignment of mitochondrial DNA from *C. nasus* and closely related species was performed using ClustalW in MEGA 11. To establish a rapid and specific detection method, we designed an RPA-LFD assay targeting a specific region of the cytochrome c oxidase subunit I (COI) gene. This assay utilizes a species-specific primer and probe set, incorporating a previously reported SNP marker to reliably differentiate *C. nasus* from *C. brachygnathus* and other non-target species [20]. Species-specific primers and probes were designed to discriminate *Coilia* species from non-target fish species.

Endpoint PCR amplification was performed on *C. nasus* and *C. brachygnathus* samples using the primers Cn-Cb-F (5′-CGCTACGCTGCTGACTGCAC-3′) and Cn-Cb-R (5′-CAGCGTGGTCTGCTGGTTCAT-3′) [20]. PCR products were analyzed by 1.5% agarose gel electrophoresis and subjected to Sanger sequencing by Tsingke Biotechnology (China). All sequence results of *C. nasus* and *C. brachygnathus* were aligned against the *C. nasus* reference genome (GenBank accession: GCA_027475355.1) in MEGA 11 to verify SNP conservation and specificity.

RPA primers/probes targeting *Coilia* species and qPCR primers/MGB probes for discriminating *C. nasus* from *C. brachygnathus* were designed using Primer Express 3.0 (Applied Biosystems, Waltham, MA, USA) (Figure 1). The species specificity of these primers and probes was further validated via BLAST analysis (BLAST+ 2.17.0). All primers and probes designed in this study—including the MGB probes labeled with FAM and VIC as 5′-reporter dyes (Table 2)—were synthesized by Beijing Liuhe BGI Technology Co., Ltd, Beijing, China. 

### 2.4. Development and Optimization of the RPA-LFD Assay

The nucleic acid amplification-lateral flow detection platform was established using the colloidal gold-based isothermal amplification kit (Amplification Future (Changzhou) Biotechnology Co., Ltd., Changzhou, China). The 50 μL reaction system was precisely formulated with 29.4 μL rehydration buffer A, 11.4 μL molecular-grade H_2_O, 2 μL primer pairs (10 μM each), and 0.6 μL nfo probe (10 μM). Following the precise addition of template DNA (2 μL), the mixture was supplemented with 2.5 μL magnesium acetate activator (Buffer B, 280 mM magnesium acetate) in 0.2 mL polypropylene tubes containing lyophilized enzyme complexes. Three thermal optimization groups (37 °C, 39 °C, and 41 °C ± 0.5 °C) were established, with incubation for 10–15 min in precision water baths after vortex homogenization (5 s, 2000 rpm). Amplification products were visualized via lateral flow chromatography using a dual-line detection cassette: a test line (TL) for specific FAM-labeled amplicons and a control line (CL) with a biotin-streptavidin conjugate, enabling naked-eye interpretation within 5 min.

### 2.5. Establishment of Duplex TaqMan-MGB qPCR Assay and Standard Curves

The 25 μL reaction system from the kit (Takara, RR390A) was used, consisting of 12.5 μL 2×TaqMan qPCR Probe Master Mix, 1 μL each of probes Cb-p-FAM and Cn-p-VIC (10 μmol/L), 0.5 μL each of forward and reverse primers (10 μmol/L), and 2 μL template DNA. The volume was adjusted to 25 μL with ddH_2_O. The reaction conditions were as follows: pre-denaturation at 95 °C for 30 s; 40 cycles of denaturation at 95 °C for 5 s and annealing at 57 °C for 31 s. Standard plasmids were subjected to 10-fold serial dilution and then amplified using a qPCR instrument according to the program. Three replicates were set for each dilution gradient. A standard curve was automatically generated by the real-time fluorescence quantitative PCR instrument, with Ct values as the *Y*-axis and the copy number of the standard as the *X*-axis.

### 2.6. Preparation of Plasmid DNA Standards

Genomic DNA from homozygous individuals of *Coilia brachygnathus* (genetically fixed for the T allele) and *Coilia nasus* (genetically fixed for the C allele) was used as the template for PCR amplification. Amplification was performed using the forward primer Cn-Cb-F: 5′-CGCTACGCTGCTGACTGCAC-3′ and reverse primer Cn-Cb-R: 5′-CAGCGTGGTCTGCTGGTTCAT-3′, targeting the sequence in GenBank accession CM050218.1. The resulting PCR products were gel-purified and cloned into the pUC57 vector. Recombinant plasmids were sent to Tsingke Biotechnology Co., Ltd. (Beijing, China) for sequencing verification. Successfully sequenced positive plasmids were designated as pUC57-Cb-T and pUC57-Cb-C, serving as standard reference materials for detecting *C. brachygnathus* and *C. nasus* samples, respectively. Plasmid standard concentrations were measured using a microvolume UV spectrophotometer, and copy numbers were calculated using the formula: Plasmid copy number (copies/μL) = [Plasmid concentration (μg/μL) × 6.02 × 10^23^ (copies/mol)]/[Plasmid size (bp) × 660 (g/mol/bp)]. Subsequently, 10-fold serial dilutions of the plasmid standards were prepared to achieve concentrations ranging from 1 × 10^8^ to 1 × 10^1^ copies/µL. During each dilution step, samples were vortex-mixed for 30 s and pipetted up and down 30 times; this mixing procedure was repeated three times to ensure thorough resuspension of the plasmid in ddH_2_O. Diluted standards were appropriately labeled and stored at −20 °C for future use.

### 2.7. Specificity Testing

This study employed the pre-established RPA-LFD and duplex TaqMan-MGB qPCR platforms to systematically evaluate target recognition specificity. Specifically, cross-reactivity experiments were performed using DNA samples from various non-target fish species—with *C. nasus* and *C. brachygnathus* DNA as positive controls—to validate the specificity of the RPA-LFD assay. Additionally, the duplex TaqMan-MGB qPCR assay was conducted to verify its ability to discriminate between *C. nasus* and *C. brachygnathus*. These experiments collectively assessed the target sequence recognition capability of the detection systems.

### 2.8. Limit of Detection (LOD) Determination

To determine the limit of detection (LOD) for the developed RPA-LFD and duplex TaqMan-MGB qPCR assays, the following analyses were conducted: 1. RPA-LFD: Serial dilutions of *C. nasus* genomic DNA, ranging from 2.35 × 10^1^ ng/µL to 2.35 × 10^−4^ ng/µL, were analyzed. 2. Duplex TaqMan-MGB qPCR: Serial dilutions of positive control plasmids pUC57-Cb-T (containing the *C. brachygnathus* target sequence) and pUC57-Cb-C (containing the *C. nasus* target sequence), ranging from 10^8^ to 10^1^ copies/µL, were tested.

### 2.9. Validation with Samples of Known Origin

A total of 171 whole fish specimens—comprising 29 *C. nasus*, 6 *C. brachygnathus*, and 136 individuals from closely related non-target species—were screened using the RPA-LFD assay. Additionally, a blinded evaluation was conducted using the developed duplex TaqMan-MGB qPCR test on 29 *C. nasus* and 6 *C. brachygnathus* dorsal muscle tissue samples routinely collected by our laboratory to verify the effectiveness of the established procedures.

## 3. Results

### 3.1. Validation of Primers and SNP Genotyping

Multiple sequence alignment of *C. nasus* and congeneric mitochondrial sequences (retrieved from NCBI) identified a conserved region within the COI gene as the target for RPA-LFD detection. RPA produced a single, distinct band for both *C. nasus* and *C. brachygnathus* (Figure 2), confirming primer efficiency. Endpoint PCR using previously published primers Cn-Cb-F/R targeted a diagnostic SNP site differentiating *C. nasus* from *C. brachygnathus* [20]. Sequencing and alignment to the *C. nasus* reference genome mapped the SNP to position 3,798,155 (C/T) in gene ENSCNAG00005025661 (chromosome 21), confirming a fixed interspecific difference (Figure 3A,B). The *C. brachygnathus* population was fixed for the T allele, while the *C. nasus* population was fixed for the C allele, matching the reference genome (Figure 3C,D). This SNP therefore reliably discriminates *C. nasus* from *C. brachygnathus*.

### 3.2. Development and Optimization of the RPA-LFD Assay

To optimize the RPA-LFD assay, we first evaluated three reaction temperatures (37 °C, 39 °C, and 41 °C). The test line exhibited the strongest signal intensity at 37 °C, with progressively weaker signals observed at higher temperatures (Figure 4A). Consequently, 37 °C was selected as the optimal reaction temperature.

We next assessed incubation times (5, 8, 10, 12, and 15 min) at 37 °C. Test line signal intensity increased progressively with longer incubation times (Figure 4B). Although a detectable band emerged at 8 min and maximal intensity was observed at 15 min, prolonged incubation increased non-specific background signals. To prevent false positives while maintaining sufficient sensitivity, the optimal reaction time was determined to be 10 min. Consequently, the diagnostically optimal conditions for the RPA-LFD assay were established as 37 °C for 10 min.

### 3.3. Specificity and Sensitivity Analysis of the RPA-LFD Assay

Specificity Assessment: The RPA-LFD assay was evaluated using DNA from *C. nasus*, *C. brachygnathus*, and samples representing 14 non-target fish species (16 samples in total). The assay exhibited high specificity, producing distinct test (T) and control (C) lines exclusively for *C. nasus* and *C. brachygnathus* samples. Critically, no T lines were observed in any of the non-target species samples (Figure 5).

Sensitivity Assessment: Detection sensitivity was determined using serial dilutions of *C. nasus* muscle tissue DNA (concentration range: 2.35 × 10^1^ to 2.35 × 10^−4^ ng/μL). The assay reliably detected the target sequence with a minimum limit of detection (LOD) of 2.35 × 10^−2^ ng/μL (23.5 pg/μL) (Figure 6), demonstrating sufficient sensitivity for practical applications.

### 3.4. Establishment of the Duplex TaqMan-MGB qPCR Standard Curve

Following quantification of the standard plasmids pUC57-Cb-T (*C. brachygnathus*) and pUC57-Cn-C (*C. nasus*), their copy numbers were calculated as 1.25 × 10^10^ copies/μL. Ten-fold serial dilutions (ranging from 10^8^ to 10^0^ copies/μL) were analyzed using TaqMan-MGB qPCR. Amplification curves were generated from the qPCR data, and standard curves were constructed with quantification cycle (Cq) values as the ordinate and log_10_(copy number) as the abscissa. Results showed amplification signals in the FAM channel when pUC57-Cb-T was used as the template (Figure 7A), while amplification signals in the VIC channel were observed with the pUC57-Cn-C template (Figure 7C). The standard curves exhibited slopes of −3.42 (FAM) and −3.35 (VIC), with correlation coefficients (R^2^) > 0.99 for both. Amplification efficiencies were 96% and 99%, respectively. These parameters demonstrate excellent linear relationships between log_10_ (copy number) and Cq values for all standard templates (Figure 7B,D).

### 3.5. Sensitivity Analysis of the Duplex TaqMan-MGB qPCR Assay

Using 10-fold serially diluted pUC57-Cb-T and pUC57-Cn-C plasmid DNA (10^4^ to 10^0^ copies/μL) as templates, the five gradient templates were amplified using the duplex TaqMan-MGB qPCR method. The results (Figure 8) show that the minimum limit of detection (LOD) for *C. brachygnathus* (FAM channel) was 10^1^ copies/reaction, and for *C. nasus* (VIC channel) was 10^2^ copies/reaction.

### 3.6. Specificity Analysis of the Duplex TaqMan-MGB qPCR Assay

The results of duplex TaqMan-MGB qPCR detection for known *C. brachygnathus* and *C. nasus* are shown in Figure 9. For *C. brachygnathus*, Ct values in the FAM channel appeared significantly earlier than those in the VIC channel (ΔCt ≥ 1) with strong FAM signals. In contrast, *C. nasus* exhibited amplification exclusively on the VIC channel, with no FAM signals detected. No amplification was observed in the negative control (H_2_O), demonstrating the excellent specificity of this method.

### 3.7. Validation Using Samples of Known Origin

Validation with Biological Samples: Muscle samples of confirmed origin were analyzed using a dual detection strategy: RPA-LFD for initial *Coilia* genus screening, followed by TaqMan-MGB qPCR for species differentiation (*C. nasus* vs. *C. brachygnathus*). The sample set comprised 6 *C. brachygnathus*, 29 *C. nasus* (Yangtze River basin), and 14 non-*Coilia* fish, totaling 50 samples.

RPA-LFD accurately identified all *Coilia* samples. Subsequent qPCR analysis showed complete concordance with the expected genotypes: all *C. brachygnathus* samples exhibited exclusive T-allele signals (FAM channel), while all *C. nasus* samples showed exclusive C-allele signals (VIC channel) (Table 3). No false positives or negatives were observed, resulting in a 100% detection rate. Negative controls (H_2_O) showed no amplification, confirming the assay’s specificity and reliability under diagnostic conditions.

## 4. Discussion

This study successfully established and validated a novel dual-platform approach integrating RPA-LFD for rapid on-site screening and TaqMan qPCR for confirmatory laboratory genotyping, enabling accurate and efficient discrimination of *C. brachygnathus* and *C. nasus*. Our results demonstrate that this integrated strategy bridges a critical methodological gap: it simultaneously addresses the urgent need for field-deployable diagnostics and the requirement for laboratory-grade precision in species authentication [13,21,22,23]. This capability is particularly vital for high-value, morphologically similar congeners facing escalating conservation threats and commercial mislabeling. Precise species identification is fundamental to combating illegal wildlife trade and fisheries fraud in the Yangtze River basin. The validated assays developed herein provide a robust technical foundation for enforcing fishing bans, safeguarding endemic biodiversity, and enhancing sustainable management of aquatic biological resources [24,25,26].

Morphological characteristics remain the mainstay of conventional fish species identification. However, biological morphology represents a phenotypic ensemble influenced by both environmental and genetic factors [27], leading to inherent subjectivity in taxonomic classification. A notable example is *C. nasus*, whose morphological taxonomy has long been controversial due to inconsistent standards: measurement biases in traits such as maxillary length or anal fin ray counts may distort species delineation [28,29]. Although researchers have attempted to improve accuracy by integrating multivariate morphometrics (e.g., principal component analysis of body length/height ratios) with traditional morphology [30], these efforts remain constrained by methodological bottlenecks: high-precision instrumentation, complex statistical software, and prolonged processing times (up to weeks). These challenges highlight the urgent need for objective, DNA-based identification methods.

DNA-based techniques, including molecular markers, random amplified polymorphic DNA (RAPD) analysis, and DNA barcoding, have been widely applied for precise species identification and genetic analysis in aquatic organisms. For instance, Cheng et al. [20] developed SNP markers (Ct-Cn_wtap and Ct-Cn_eif2b4) that distinguish *C. brachygnathus*, *C. nasus taihuensis*, and *C. nasus* with 100% accuracy, providing an efficient tool for species identification and conservation of *Coilia* species in the Yangtze River basin.

DNA barcoding and metabarcoding have proven highly effective in detecting species mislabeling. Wang et al. [31] reported a 50% mislabeling rate (16/32 samples) in Chinese commercial salmon products, including undeclared substitutions with low-value fish and non-fish species. Furthermore, Rasmussen et al. [32] demonstrated that COI gene barcoding reliably differentiates all seven commercially significant North American salmonid species, exhibiting a 32-fold greater mean congeneric divergence (8.22%) than intraspecific variation (0.26%), confirming its utility for commercial product authentication. Similarly, Partis and Wells [33] established that RAPD analysis generates species-specific DNA profiles for eight commercially important fish species, enabling detection of mislabeled products. However, conventional DNA-based methods such as PCR-based DNA barcoding, RAPD (random amplified polymorphic DNA), and other DNA molecular markers, while highly accurate, suffer from limitations including complex procedures, time-consuming processes, reliance on specialized laboratory instruments (thermocyclers for amplification and sequencers), and higher costs. Therefore, the development and application of novel rapid on-site molecular detection technologies are imperative.

To address the limitations of morphological and PCR-based methods, emerging isothermal amplification technologies—such as RPA (recombinase polymerase amplification) [34], LAMP (loop-mediated isothermal amplification) [35], and CRISPR/Cas12 [36]—offer significant advantages. First, these techniques drastically reduce reaction times to 20–40 min by amplifying nucleic acids under isothermal conditions, eliminating the need for intricate thermal cycling [37]. Second, the use of specific primers ensures efficient amplification of target DNA fragments, guaranteeing excellent detection accuracy. More importantly, these methods significantly lower technical barriers: they are easy to operate, require no specialized expertise or complex laboratory equipment, and are thus ideal for on-site rapid testing [38]. Their broad sample compatibility also enhances efficiency in forensic applications and species identification. Among various isothermal amplification methods, RPA stands out due to its unique advantages. First, it simplifies reaction setup, requiring only two primers and a single labeled probe. Second, core enzymes and related reagents are now fully domestically produced in China, reducing costs and facilitating widespread application in resource-constrained settings [17,18,39].

Therefore, leveraging RPA-LFD technology, we first differentiated *Coilia* species from those of other genera. Building on this, we developed a duplex TaqMan-MGB qPCR assay targeting single nucleotide polymorphism (SNP) sites to distinguish between *C. brachygnathus* and *C. nasus* within the *Coilia* genus. This assay directly identifies sequence variations via distinct fluorescent signals, thereby achieving accurate differentiation of *C. nasus* from *C. brachygnathus*. The duplex TaqMan-MGB qPCR technique further enhances discrimination accuracy and specificity. It employs species-specific probes designed for diagnostic SNP sites (C/T) unique to *C. brachygnathus* and *C. nasus*, enabling precise species differentiation through differential fluorescent signals. The integration of TaqMan-MGB probes not only increases binding stability between the probe and target sequence but also significantly reduces background noise from non-specific amplification, thereby improving detection sensitivity and reliability. Additionally, the duplex qPCR design allows simultaneous detection of both species in a single reaction, streamlining experimental workflows and reducing reagent consumption.

First, this study successfully established a rapid DNA-based species identification technique for *Coilia* in the Yangtze River using the RPA-LFD assay. By targeting the conserved region of the COI gene in *Coilia*, we designed RPA-specific primers and probes. Through systematic optimization of reaction conditions and amplification systems, the specificity and sensitivity of the assay were significantly enhanced. Leveraging the technical advantages of RPA, a lateral flow dipstick RPA assay was developed, yielding results within 10 min with a detection limit of 2.35 × 10^−2^ ng/μL. Validation testing confirmed its excellent performance: specificity tests demonstrated strong specificity with no cross-reactivity with DNA from closely related species. Validation using 149 known fish samples showed 100% concordance with expected results. These findings clearly indicate that the established RPA-based assay provides a reliable tool for field applications, offering high efficiency and accuracy for rapid *Coilia* identification. Second, targeting a diagnostic SNP (chr21:3798155, C/T) that stably differentiates *C. brachygnathus* and *C. nasus* within the *Coilia* genus, this study designed dual-labeled (FAM/VIC) MGB probes to develop a duplex TaqMan-MGB qPCR assay. Experimental results showed that this method enables differentiation of the two species in a single tube: For *C. brachygnathus*, Ct values in the FAM channel were significantly earlier than those in the VIC channel (ΔCt ≥ 1), with a FAM detection limit of 125 copies/reaction. For *C. nasus*, amplification occurred exclusively in the VIC channel, with a detection limit of 12.5 copies/μL. These results indicate that the developed method facilitates rapid, large-scale screening within the *Coilia* genus, enabling precise discrimination between *C. brachygnathus* and *C. nasus*.

RPA, developed by Piepenburg et al. (2006) [16], is widely used for rapid pathogen detection in animal [40] and plant [41] systems, achieving sensitivities as low as 2.5 copies/μL in clinical point-of-care testing (POCT, e.g., SARS-CoV-2) [42]. While our RPA-LFD assay for *C. nasus* exhibits 4-fold lower sensitivity, this aligns with species identification priorities that favor specificity and field deployability over ultra-trace detection. Future optimization of primers and probes could enhance sensitivity for trace samples (e.g., environmental DNA [eDNA] or degraded tissue) [43]. TaqMan-MGB qPCR provided confirmatory genotyping with superior specificity. MGB probes enabled precise discrimination of conserved SNPs between *C. nasus* and *C. brachygnathus* via enhanced hybridization stability [44]. Its higher tolerance to inhibitors and quantitative accuracy [45] proved critical for validating field results and processed samples, solidifying its role in the dual-platform framework.

The core innovation of this study lies in its multi-tiered integration strategy. This dual-platform approach synergistically combines the speed, simplicity, and field deployability of RPA-LFD for initial screening with the high specificity, sensitivity, and confirmatory capability of duplex TaqMan qPCR for definitive genotyping. Compared to traditional morphological identification—characterized by subjectivity and time consumption—our molecular method delivers objective results rapidly (minutes/hours versus days/weeks), effectively circumventing the risk of misidentification arising from phenotypic plasticity. Furthermore, in contrast to reliance on a single molecular technique (e.g., standalone SNP analysis or DNA barcoding), this integrated strategy significantly enhances the overall robustness and reliability of species discrimination. This advantage is particularly pronounced for closely related species or potentially degraded samples encountered in real-world scenarios.

This methodology holds substantial promise for diverse applications. In Yangtze River Basin fishery surveys, the RPA-LFD component enables rapid, on-site identification during field sampling, while the qPCR assay provides high-confidence data to accurately monitor *C. brachygnathus* and *C. nasus* population dynamics—supporting science-based conservation and management policies. For law enforcement efforts to combat illegal fishing, the RPA-LFD offers a tool for near-real-time screening of suspicious catches at checkpoints; positive results can be rapidly confirmed in laboratories via qPCR, generating forensically robust evidence. In ecological research, the precise discrimination capability facilitates detailed investigations into niche differentiation, interspecific interactions, and responses to environmental stressors between these sympatric species.

In summary, we have established and validated an integrated dual-platform strategy that effectively bridges the gap between rapid field screening and high-accuracy laboratory confirmation for discriminating *C. brachygnathus* and *C. nasus*. By innovatively combining RPA-LFD for efficient preliminary identification with a highly specific duplex TaqMan-MGB qPCR assay targeting a diagnostic SNP, this approach delivers a solution that is both operationally practical and scientifically rigorous. Its demonstrated sensitivity, specificity, speed, and layered design underscore its significant practical value for advancing research, conservation management, fisheries monitoring, and legal enforcement efforts concerning these ecologically critical Yangtze River species.

## Figures and Tables

**Figure 1 foods-14-03484-f001:**
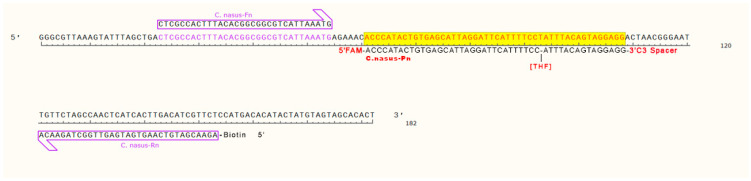
Design of RPA primers and probes for *Coilia*.

**Figure 2 foods-14-03484-f002:**
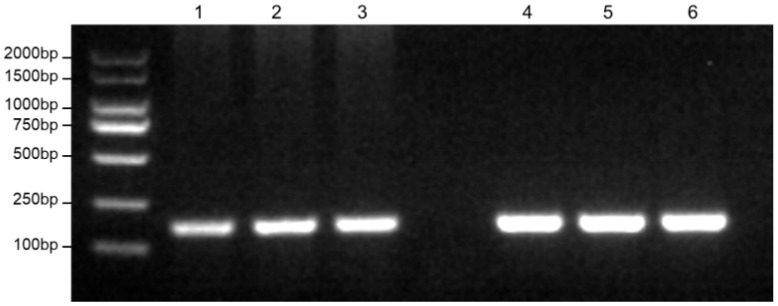
Results of RPA primer efficiency verification. 1–3: *C. nasus*; 4–6: *C. brachygnathus*.

**Figure 3 foods-14-03484-f003:**
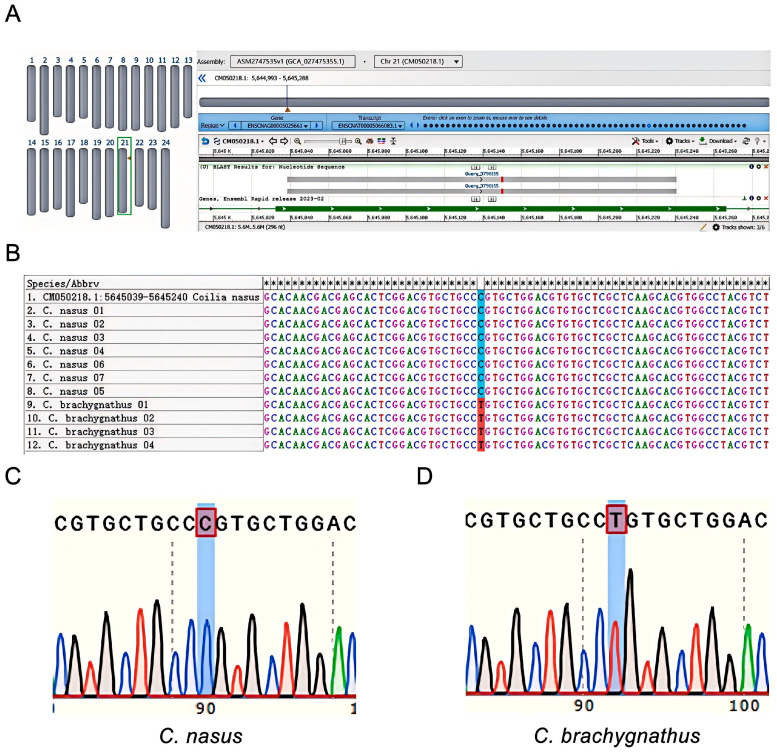
SNP verification results. (**A**,**B**) Alignment of sequencing results against the *C. nasus* reference genome (GCA_027475355.1), revealing localization of primer-amplified sequences from *C. nasus* and *C. brachygnathus* samples to position 3,798,155 within gene ENSCNAG00005025661 (CM050218.1: chr21). (**C**) Sequencing results for *C. nasus* samples, demonstrating fixation of the C allele. (**D**) Detection results for *C. brachygnathus* samples, showing fixation of the T allele.

**Figure 4 foods-14-03484-f004:**
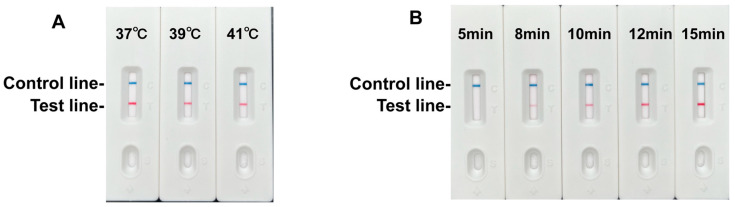
Optimization of RPA LFD assay. (**A**) RPA-LFD assays were performed at the indicated temperatures. (**B**) The test line is clearly visible at 37 °C when the amplification time exceeds 10 min.

**Figure 5 foods-14-03484-f005:**
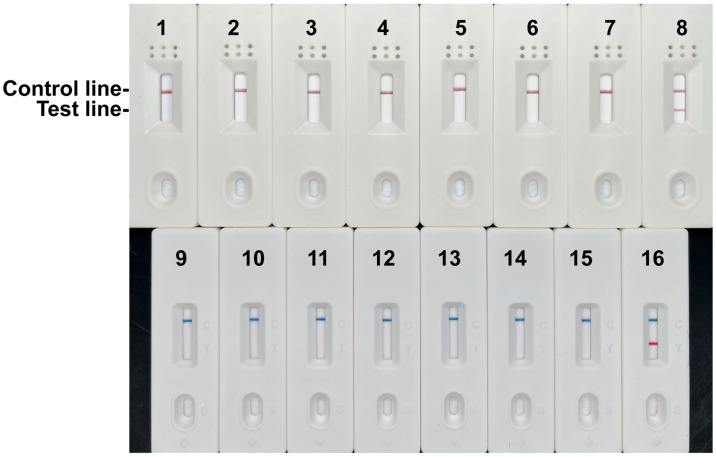
Specificity results of the RPA-LFD assay. 1, *Tachysurus fulvidraco*; 2, *Pelteobagrus eupogon*; 3, *Pelteobagrus vachellii*; 4, *Pelteobagrus nitidus*; 5, *Hypophthalmichthys molitrix*; 6, *Hypophthalmichthys nobilis*; 7, *Mylopharyngodon piceus*; 8, *Coilia nasus*; 9, *Ctenopharyngodon idella*; 10, *Siniperca chuatsi*; 11, *Siniperca knerii*; 12, *Culter alburnus*; 13, *Chanodichthys dabryi*; 14, *Chanodichthys mongolicus*; 15, *Culter erythropterus*; 16, *Coilia brachygnathus*.

**Figure 6 foods-14-03484-f006:**
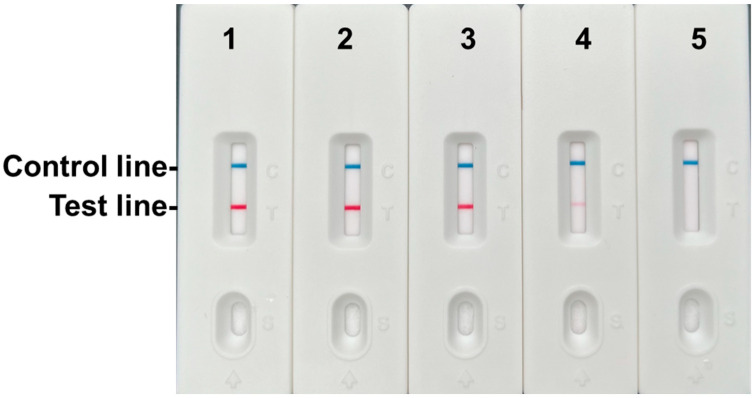
Sensitivity results of the RPA-LFD assay. 1–5: 2.35 ng/μL, 0.235 ng/μL, 0.0235 ng/μL, 0.00235 ng/μL, 0.000235 ng/μL.

**Figure 7 foods-14-03484-f007:**
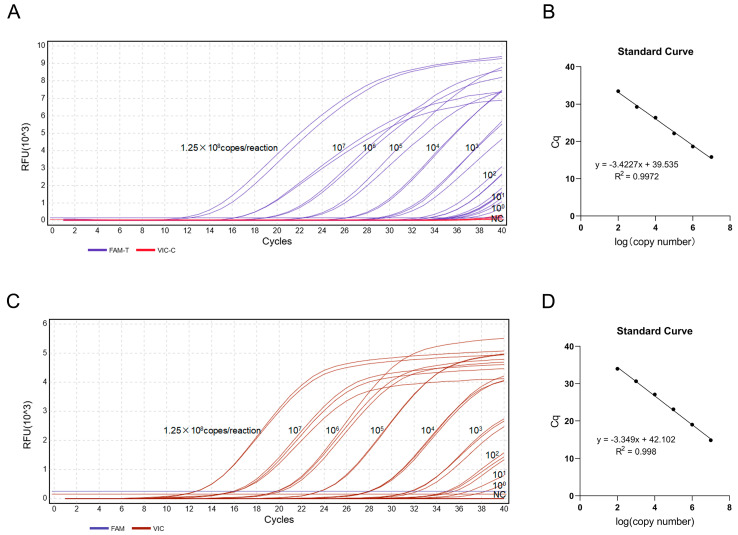
Standard curves and amplification plots for duplex TaqMan-MGB qPCR. (**A**) Amplification plot (FAM channel) for *C. nasus* (pUC57-Cn-T plasmid); (**B**) Standard curve (FAM channel) for *C. nasus*. (**C**) Amplification plot (VIC channel) for *C. brachygnathus* (pUC57-Cb-C plasmid). (**D**) Standard curve (VIC channel) for *C. brachygnathus*.

**Figure 8 foods-14-03484-f008:**
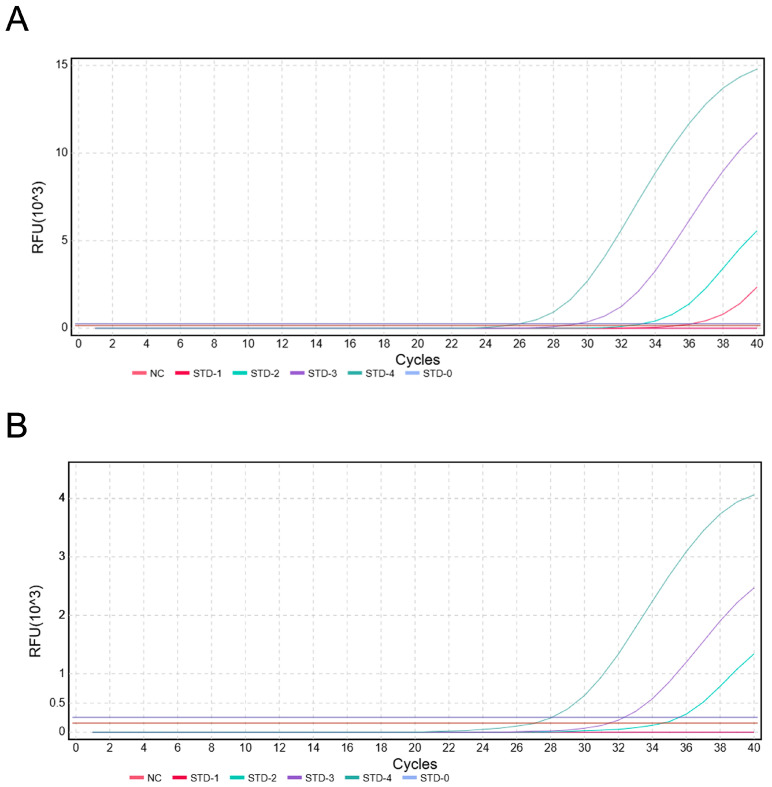
The results of duplex TaqMan-MGB qPCR sensitivity (**A**) Limit of detection (LOD) for FAM (*C. brachygnathus*): 10^1^ copies/reaction. (**B**) Limit of detection (LOD) for VIC (*C. nasus*): 10^2^ copies/reaction.

**Figure 9 foods-14-03484-f009:**
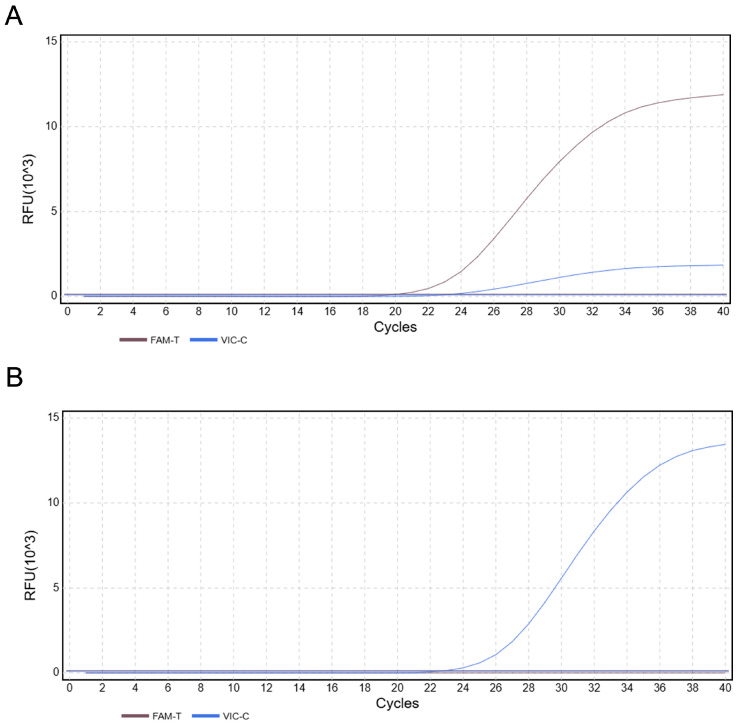
Specificity results of duplex TaqMan-MGB qPCR. (**A**) Specificity results for *C. brachygnathus* (**B**) Specificity results for *C. nasus*.

**Table 1 foods-14-03484-t001:** Species composition and ecological characteristics of experimental samples.

Species	Sample Sizes	Habitat Vegetation Type
*Coilia nasus*	29	Migratory (freshwater—brackish water) upstream during the breeding season, endangered species
*Coilia brachygnathus*	6	Small freshwater/brackish water (some migratory) fish, commonly found in estuaries
*Tachysurus fulvidraco*	14	Fresh water at the bottom, still water/slow flow
*Pelteobagrus eupogon*	6	Freshwater bottom layer, rivers/lakes
*Pelteobagrus vachellii*	6	Freshwater bottom, slow-flowing/reservoir
*Pelteobaggrus nitidus*	6	Freshwater bottom, stream/estuary
*Hypophthalmichthys molitrix*	20	The upper and middle layers of freshwater, open water areas
*Hypophthalmichthys Nobilis*	14	Upper and middle layers of freshwater, lakes/reservoirs
*Mylopharyngodon piceus*	6	Freshwater bottom, deep water area/river
*Ctenopharyngodon Idella*	12	The middle and lower layers of freshwater, an area rich in aquatic plants
*Siniperca chuatsi*	14	Freshwater bottom layer, crevices in rocks/water grass; Ambush carnivorous, high economic value
*Siniperca knerii*	6	Freshwater bottom layer, clear stream; Large eyes can adapt to weak light and are widely distributed
*Culter alburnus*	14	Upper and middle layers of freshwater, open water area; Fierce and predatory, good at jumping
*Chanodichthys dabryi*	6	Upper and middle layers of freshwater, slow-flowing/lake; Small fish, moving in groups
*Chanodichthys mongolicus*	6	The upper and middle layers of freshwater, rivers/lakes
*Culter erythropterus*	6	The upper and middle layers of freshwater, the edges of aquatic plants

**Table 2 foods-14-03484-t002:** The primers and probes for RPA-LFD and TaqMan-MGB qPCR.

Name	Sequence 5′–3′	Product Size (bp)	Reference
*C. nasus*-Fe	CTCGCCACTTTACACGGCGGCGTCATTAAATG	103	Designed in this study
*C. nasus*-Rn	5′Biotin-AGAACGATGTCAAGTGATGAGTTGGCTAGAACA		
*C. nasus*-Pn	5′FAM-ACCCATACTGTGAGCATTAGGATTCATTTTCC[THF]ATTTACAGTAGGAGG-3′C3 Spacer		
Cn-Cb-F	CGCTACGCTGCTGACTGCAC	202	[20]
Cn-Cb-R	CAGCGTGGTCTGCTGGTTCAT		
Cb-p-FAM	5′FAM-AGCACAGGCAGCA-3′MGB		Designed in this study
Cn-p-VIC	5′VIC-AGCACGGGCAGCA-3′MGB		

**Table 3 foods-14-03484-t003:** Results of practical sample detection.

Number of Samples	Known Sample Name	Detection Method			Identify
		RPA-LFD	Duplex Taqman-MGB qPCR		
		*Coilia* (+) non-*Coilia* (−)	FAM-T	VIC-C	
1	*C. brachygnathus*	+	29.98	32.12	*C. brachygnathus*
2	*C. brachygnathus*	+	22.08	23.72	*C. brachygnathus*
3	*C. brachygnathus*	+	22.03	23.87	*C. brachygnathus*
4	*C. brachygnathus*	+	21.28	22.90	*C. brachygnathus*
5	*C. brachygnathus*	+	23.50	NA	*C. brachygnathus*
6	*C. brachygnathus*	+	23.37	24.45	*C. brachygnathus*
7	*C. nasus*	+	NA	29.15	*C. nasus*
8	*C. nasus*	+	NA	23.37	*C. nasus*
9	*C. nasus*	+	NA	24.81	*C. nasus*
10	*C. nasus*	+	NA	27.41	*C. nasus*
11	*C. nasus*	+	NA	24.48	*C. nasus*
12	*C. nasus*	+	NA	32.46	*C. nasus*
13	*C. nasus*	+	NA	24.19	*C. nasus*
14	*C. nasus*	+	NA	24.57	*C. nasus*
15	*C. nasus*	+	NA	24.86	*C. nasus*
16	*C. nasus*	+	NA	23.83	*C. nasus*
17	*C. nasus*	+	NA	25.27	*C. nasus*
18	*C. nasus*	+	NA	25.43	*C. nasus*
19	*C. nasus*	+	NA	22.08	*C. nasus*
20	*C. nasus*	+	NA	22.20	*C. nasus*
21	*C. nasus*	+	NA	24.46	*C. nasus*
22	*C. nasus*	+	NA	23.19	*C. nasus*
23	*C. nasus*	+	NA	25.10	*C. nasus*
24	*C. nasus*	+	NA	24.53	*C. nasus*
25	*C. nasus*	+	NA	23.56	*C. nasus*
26	*C. nasus*	+	NA	22.94	*C. nasus*
27	*C. nasus*	+	NA	22.11	*C. nasus*
28	*C. nasus*	+	NA	21.12	*C. nasus*
29	*C. nasus*	+	NA	22.99	*C. nasus*
30	*C. nasus*	+	NA	23.91	*C. nasus*
31	*C. nasus*	+	NA	24.06	*C. nasus*
32	*C. nasus*	+	NA	22.79	*C. nasus*
33	*C. nasus*	+	NA	23.63	*C. nasus*
34	*C. nasus*	+	NA	22.53	*C. nasus*
35	*C. nasus*	+	NA	23.74	*C. nasus*
36	*Tachysurus fulvidraco*	−	/	/	non-*Coilia*
37	*Pelteobagrus eupogon*	−	/	/	non-*Coilia*
38	*Pelteobagrus vachellii*	−	/	/	non-*Coilia*
39	*Pelteobaggrus nitidus*	−	/	/	non-*Coilia*
40	*Hypophthalmichthys molitrix*	−	/	/	non-*Coilia*
41	*Hypophthalmichthys Nobilis*	−	/	/	non-*Coilia*
42	*Mylopharyngodon piceus*	−	/	/	non-*Coilia*
43	*Ctenopharyngodon idella*	−	/	/	non-*Coilia*
44	*Siniperca chuatsi*	−	/	/	non-*Coilia*
45	*Siniperca knerii*	−	/	/	non-*Coilia*
46	*Culter alburnus*	−	/	/	non-*Coilia*
47	*Chanodichthys dabryi*	−	/	/	non-*Coilia*
48	*Chanodichthys mongolicus*	−	/	/	non-*Coilia*
49	*Culter erythropterus*	−	/	/	non-*Coilia*
50	H_2_O	−	-	-	NA

“NA” represents Not Available.

## Data Availability

The original contributions presented in this study are included in the article. Further inquiries can be directed to the corresponding authors.

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
