# Peer review of "Integrating RPA-LFD and TaqMan qPCR for Rapid On-Site Screening and Accurate Laboratory Identification of *Coilia brachygnathus* and *Coilia nasus* in the Yangtze River"

_foods, 2025, doi:10.3390/foods14203484_

Round 1
Reviewer 1 Report
Comments and Suggestions for Authors
5th September 2025
Manuscript ID: foods-3844487
Type of manuscript: Article
Title: Integrating RPA-LFD and TaqMan qPCR for Rapid On-Site Screening and Accurate Laboratory Identification of Coilia brachygnathus and Coilia nasus in the Yangtze River
Authors: Yu Lin, Suyan Wang, Min Zhang, Na Wang, Hongli Jing, Jizhou Lv, Shaoqiang Wu.
The work entitled “Integrating RPA-LFD and TaqMan qPCR for Rapid On-Site Screening and Accurate Laboratory Identification of Coilia brachygnathus and Coilia nasus in the Yangtze River” concerns development of the two complementary methods for fish identification i.e. RPA-LFD and TaqMan qPCR. RPA-LFD was used for Coilia species identification and the results of this method were received, with the application of the proper reagents, after 10 minutes samples incubation in 37 °C. In this method the primers and the probe designed on the base of COI were applied. Next the endpoint PCR and Sanger sequencing were made. On the base of the received sequences, SNP conservation and specificity were verified. The last step of the experiment was developing a dual TaqMan MGB qPCR. As a result, the developed method made possible to distinguish the Coilia brachygnathus and Coilia nasus in a one reaction tube. In a case of Coilia brachygnathus Ct values in the FAM channel appeared significantly earlier than those in the VIC channel with the detection limit of 125 copies per reaction. For Coilia nasus only the VIC channel amplification was observed with the detection limit 12.5 copies per reaction. The validation of the developed method confirmed 100 % consistency with the 171 known tissue samples.
The presented study is very interesting and certainly needed, as the authors indicated, for the fishery management, conservation and fight against food market adulteration. A particularly interesting solution was elaboration two complementary methods, one (RPA-LFD) which allows for the rapid on-site identification of fish belonging to the Colia species, and the other (TaqMan qPCR) permitting for the precise differentiation of Colia nasus from Colia brachygnathus in laboratory conditions.
This work fits the objectives and scope of the journal “Foods”. The literature was selected properly and over 73 % quoted works came from the last 10 years. The research was carried out properly and appropriate analyses were applied. Both the methodology and the research results have been described correctly but some minor errors appeared.
- Page 4, line 18, section 2.3., the names of the primers are ”Cn-Cb-F”, “Cn-Cb-R” but in the page 5, table 2; page 5, section 2.6; page 6, section 3. Results, 3.1 etc. their names are different i.e ”Cb-Cn-F”, “Cb-Cn-R”. It looks like lack of the consequence in nomenclature.
- Page 4, figure 1, the description of the figure is not clear. What mean the abbreviations: “C. nasus-Pe”, “C. nasus-Pn”, “C. nasus-Re”? Why “C. nasus-Pe” and “C. nasus-Re” are not presented in the table 2?
- The tests presented in the paper were performed only on raw samples? It would be interesting to test the elaborated method on heat-treated samples too.
- Page 2, line 5, in the sentence “However, populations of nasus have declined severely due to overfishing, habitat degradation, and river fragmentation, leading to its designation as Endangered (EN) on China's Biodiversity Red List [8, 9]” double space after word “However”.
- Page 2, line 11 “While this supports nasus conservation, it has inadvertently facilitated commercial mislabeling, where farmed C. brachygnathus substitutes for high-value C. nasus, whether unintentionally or fraudulently [10]” double space after word “high-value”.
- Page 2, line 13, “Consequently, accurate species discrimination is critical for sustaining stocks through evidence-based fisheries management, protecting consumers from economic deception, and ensuring market integrity” double space after word “sustaining”.
The above mistakes do not affect the positive opinion about the work. I will accept the submitted manuscript if the authors correct the minor errors indicated.

Reviewer 2 Report
Comments and Suggestions for Authors
his study provides a valuable contribution to species-level identification of two Coilia species, with potential relevant benefits for both fisheries management and market traceability. While the manuscript presents several noteworthy elements, certain sections may benefit from revision prior to publication. The following comments are offered in the hope that they will assist the authors in further refining the manuscript’s content. The reviewer apologizes for the seemingly lengthy comments, but in the absence of line numbers, it was necessary to reproduce entire paragraphs under review
Abstract:
The following is a proposed revision of the initial sentence :”Accurate differentiation between Coilia brachygnathus and Coilia nasus is imperative for the effective management of fisheries, the conservation of aquatic ecosystems, and the mitigation of commercial fraud.”
The incipit of the sentence “Here, we developed and validated a novel integrated approach that combines TaqMan quantitative real-time” could be rephrase as “In this study, a novel integrated approach was developed and validated, combining TaqMan quantitative real-time PCR (qPCR)..”
Introduction
With respect to the following sentence “To alleviate fishing pressure and meet market demand, C. brachygnathus (short jaw tapertail anchovy)—a morphologically similar congener—has been widely introduced for aquaculture. While this supports C. nasus conservation, it has inadvertently facilitated commercial mislabeling, where farmed C. brachygnathus substitutes for high-value C. nasus, whether unintentionally or fraudulently”:
- Consider integrating a reference to concrete management actions taken to alleviate fishing pressure on nasus, such as fewer licenses and a reduced fishing timeframe (Dai P., Yan Y., Zhu X., Tian J., Ma F., Liu K. (2020). Status of Coilia nasus resources in the National Aquatic Germplasm Resources Conservation Area in the Anqing Section of the Yangtze River. J. Fish. Sci. China 27, 1267–1276. doi:10.3724/SP.J.1118.2020.20130)
- According to FishBase, the common name associated with C. brachygnathus is Yangtse grenadier anchovy. Does the name provided by the authors reflect the English translation of the commonly used Chinese name? Kindly specify.
- the associated reference appears not supporting the assumption. It would be advisable for an alternative study to be provided.
Materials and methods
“The mitochondrial genome sequence of Coilia nasus (GenBank accession: KJ710626.2) was retrieved from NCBI (https://www.ncbi.nlm.nih.gov/). Multiple sequence alignment of mitochondrial DNA from C. nasus and closely related species was performed using ClustalW in MEGA 11.”
In general, the sentence should be revised to clearly convey to the reader the specific purpose of the sequence retrieval and the alignment described As of today, four complete mitochondrial genomes for Coilia nasus are available in NCBI. The sequence selected by the authors corresponds to a record derived from a specimen collected in the Yangtze River. Given the purpose of the sequence retrieval, it would be appropriate to clarify why the remaining sequences were not considered. Additionally, please specify the number of sequences and the accession numbers for each of the closely related species were included in the alignment.
“A specific region within the cytochrome c oxidase subunit I (COI) gene—harboring a previously reported SNP marker that stably differentiates C. nasus and C. brachygnathus—was selected as the target for RPA-LFD detection”
Please specify the position of the region within COI gene and add a reference to support the statement and specifically the mention of “a previously reported”data.
“The species specificity of these primers and probes was further validated via BLAST analysis.”
Please clarify the statement and how the specificity was verified through BLAST.
Results
“Endpoint PCR using previously published primers Cb-Cn-F/R targeted a diagnostic SNP site differentiating C. nasus from C. brachygnathus.”
In the text, the authors state that the primer pair was designed in a previous study, yet no bibliographic reference is provided to support this claim. Furthermore, this statement is apparently in contrast with the Materials and Methods section, where—unless there has been a misinterpretation—the primers seem to be included in the panel of primers and probes designed within the present study (Table 2). Could you please clarify this point?
“RPA amplification produced a single, distinct band for both C. na sus and C. brachygnathus (Fig. 2), confirming primer specificity.”
To properly assess reaction specificity, the reviewer notes the absence of amplification tests against the non-target species included in the study
